# Use of Remote Sensing Data to Improve the Efficiency of National Forest Inventories: A Case Study from the United States National Forest Inventory

**Andrew J. Lister** [1,*]**, Hans Andersen** [2]**, Tracey Frescino** [3]**, Demetrios Gatziolis** [2]**, Sean Healey** [3]**, Linda S. Heath** [4]**, Greg C. Liknes** [1]**, Ronald McRoberts** [1]**, Gretchen G. Moisen** [3]**, Mark Nelson** [1]**, Rachel Riemann** [1]**, Karen Schleeweis** [3]**, Todd A. Schroeder** [5]**, James Westfall** [1] **and B. Tyler Wilson** [1]

1   USDA Forest Service, Northern Research Station, York, PA 27410, USA; greg.liknes@usda.gov (G.C.L.); mcrob001@umn.edu (R.M.); mark.d.nelson@usda.gov (M.N.); rachel.riemann@usda.gov (R.R.); james.westfall@usda.gov (J.W.); barry.wilson@usda.gov (B.T.W.)
2   USDA Forest Service, Pacific Northwest Research Station, Portland, OR 97205, USA; hans.andersen@usda.gov (H.A.); demetrios.gatziolis@usda.gov (D.G.)
3   USDA Forest Service, Rocky Mountain Research Station, Ogden, UT 84401, USA; tracey.frescino@usda.gov (T.F.); sean.healey@usda.gov (S.H.); gretchen.g.moisen@usda.gov (G.G.M.); karen.schleeweis@usda.gov (K.S.)
4   USDA Forest Service, Washington Office, Washington, DC 20250, USA; linda.heath@usda.gov
5   USDA Forest Service, Southern Research Station, Knoxville, TN 37919, USA; todd.schroeder@usda.gov
*   Correspondence: andrew.lister@usda.gov

**Abstract:** Globally, forests are a crucial natural resource, and their sound management is critical for human and ecosystem health and well-being. Efforts to manage forests depend upon reliable data on the status of and trends in forest resources. When these data come from well-designed natural resource monitoring (NRM) systems, decision makers can make science-informed decisions. National forest inventories (NFIs) are a cornerstone of NRM systems, but require capacity and skills to implement. Efficiencies can be gained by incorporating auxiliary information derived from remote sensing (RS) into ground-based forest inventories. However, it can be difficult for countries embarking on NFI development to choose among the various RS integration options, and to develop a harmonized vision of how NFI and RS data can work together to meet monitoring needs. The NFI of the United States, which has been conducted by the USDA Forest Service's (USFS) Forest Inventory and Analysis (FIA) program for nearly a century, uses RS technology extensively. Here we review the history of the use of RS in FIA, beginning with general background on NFI, FIA, and sampling statistics, followed by a description of the evolution of RS technology usage, beginning with paper aerial photography and ending with present day applications and future directions. The goal of this review is to offer FIA's experience with NFI-RS integration as a case study for other countries wishing to improve the efficiency of their NFI programs.

**Keywords:** national forest inventory; forest monitoring; remote sensing; forest sampling; inventory efficiency; Forest Inventory and Analysis

## 1. Introduction

### 1.1. Value of National Forest Inventory (NFI) Data: Management, Research, Policy Decisions

Many nations consider natural resource monitoring (NRM) systems to be critical sources of information for making decisions on natural resource management, planning, and policy. The value

of NRM results lays in their usefulness for answering key monitoring or other science questions and making decisions. Without reliable information, management decisions can be ill-informed, leading to poor outcomes for ecosystem health and human well-being. The advancement of science also depends on NRM data; they can serve as the foundation for a diversity of both basic and applied science applications. To make investments in NRM cost-effective, it is vital that nations design highly-efficient, long-term NRM systems.

Forests cover 31%, or 4.06 billion ha, of the world's total land area [1]. An efficient National Forest Inventory (NFI) can therefore be a key component of a nation's NRM system. NFIs typically consist of a statistical sample-based system in which data on attributes relevant to stakeholders are collected on a set of NFI sampling units distributed within an area referred to as a population [2,3]. Key NFI attributes often include variables associated with trees (e.g., species, diameter, height, health status), wildlife habitat (e.g., vertical structure, standing dead trees), and the land on which the forest grows (e.g., land cover, land use, ownership, management type) [4]. It is common for NFIs to use observations based on remote sensing (RS) throughout the system, such as in data collection, estimation, and analysis and reporting [5–7].

Many countries' governments, such as that of the United States [8], mandate NFI information or otherwise provide NFI program direction. For example, recent legislation in the United States (the Agriculture Improvement Act of 2018 (P.L. 115-334, section 8632), commonly referred to as the 2018 Farm Bill), explicitly directs the USDA Forest Service (USFS) to find efficiencies in its NFI program through the use of advanced technologies such as RS and to engage other stakeholders in these efforts. The Forest Inventory and Analysis (FIA) program, which executes the NFI, has gained efficiencies and created new products for decades through careful investments in technologies like RS [9–11].

For the purposes of this review, we define RS data as those which are collected by instruments, typically mounted on an aerial or space-borne platform, that are not in direct contact with the subject of the observation. These data are often stored in the form of images or other spatially-referenced data types, and are typically produced when light or other radiation interacts with a target object and is received by a sensor mounted on the instrument. We discuss many types of RS data, including those from satellite, airborne and ground-based platforms employing passive optical sensors, as well as those from active sensors such as radio detection and ranging (radar) and light detection and ranging (lidar).

The goal of this review is to, using the FIA program's experience as a case study, provide examples of RS integration strategies for countries seeking to improve efficiency and add value to their inventories. It is not intended to be an exhaustive listing of RS activities in FIA; rather, it focuses on key work that is based on either FIA data or institutional knowledge generated by the program. Our aim is to offer FIA as a model for other NFIs by describing the statistical theory that supports efficiency gains with RS, the progression of FIA's use of RS data from its inception to modern times, and potential future directions. Our objective is to provide a better understanding of how RS technology has been and currently is integral to forest inventory science and forest management.

*1.2. Value and Uses of FIA Data*

The NFI of the United States is a well-documented case study of a multi-resource inventory that offers benefits to many types of stakeholders. Conducted by the USFS FIA Program, it has been in operation in various forms for nearly a century. It serves as the definitive source for forest resource information at the national and state scales [8,12,13]. The inventory consists of approximately 326,000 permanent, remeasured plots distributed across both forest and nonforest areas of the United States, its territories, and affiliated islands. Forest trees and certain land-type variables such as forest type and site index are only measured in portions of plots considered to be forest based on FIA's definition, and other land-type variables, such as land cover, use and, in some areas, ownership, are measured on all plots. In some urban and other areas, trees outside forest are measured on nonforest portions of plots, and on a subset of plots, additional forest health variables are measured. The base intensity is approximately one plot per 2400 ha in most areas, with plots assigned to random locations

within hexagonal cells formed from a tessellation of the population area. Varying by portion of the country, between 10 and 20% of the plots are generally sampled each year with a spatially-balanced (panelized) design. Estimates are produced using post-stratified estimation, with RS-based maps used to create strata [14,15].

The inventory is foundational for annual and 5 year reports [16,17], Renewable Resource Planning Act Assessments [13,18,19], and the U.S. Report on Sustainable Forests [20,21]. In addition, the data collected by the FIA program are used for many scientific studies. In a review of literature published or in-press between 1976 and 2001, Rudis [22,23] found over 1400 research publications that relied at least in part on FIA data. Tinkham et al. [24] reviewed a subsample of more recent (1991–2017) research publications that used FIA data. Both studies reported that FIA data contributed to a large number of science applications, including the carbon cycle, forest products and growth, climate, forest health, biological diversity, and inventory statistics [24].

In addition to science applications, FIA data have been used extensively for both local and regional land management. Hoover et al. [25] reviewed several ways that they have been used by the USFS National Forest System (NFS) for forest management, including informing land management plan revisions, assessing progress toward management plan goals, monitoring wildlife habitat, and characterizing the status of key resources. In a similar study, Wurtzebach et al. [26] identified ways FIA data are used to satisfy NFS monitoring requirements, including providing information on structure, function, composition, watershed condition and trends, carbon stocks, insect and disease mortality, wildfire effects, at-risk species, timber suitability, and other required data elements. Dugan et al. [27] and Birdsey et al. [28] used FIA data and satellite imagery and its derivatives to model the impact of various types of forest disturbance on the carbon dynamics of NFS lands. Finally, Randolph et al. [29] and Vogt and Koch [30] identified both potential and demonstrated uses of FIA data for forest health and invasive pest studies, such as research on invasive plant species, risk assessment and mapping, and monitoring impacts of invasive pests.

Other important uses of FIA data abound, and include use by the forest products industry for economic projections and processing facility siting, by non-governmental organizations for supporting their advocacy activities, and by the general public for satisfying various information and educational needs. These examples are a subset of the large number of basic and applied science applications to which FIA data and expertise have contributed [31].

### 1.3. Background on Efficiency

In the following section, we introduce background needed to help explain how FIA has traditionally used RS data to improve efficiency. Survey sampling statistical concepts and statistical efficiency are first described. This is followed by a description of ways to consider how improving statistical efficiency through RS data integration improves economic efficiency of the NFI.

### 1.4. Improvement of Statistical Efficiency—Statistical Inference

NFIs generally subscribe to the basic principles of statistical inference whereby they express their estimates in probabilistic terms [32], primarily using confidence intervals. The confidence interval width is closely related to the precision of an estimate, i.e., the shorter the confidence interval, the greater the precision. Additional expressions of precision include variances and standard errors of estimates, margins of error, sampling errors, or simply uncertainty, all of which serve as metrics that a data user will consider when making judgments or taking actions based on FIA estimates. These precision metrics are, therefore, fundamental barometers of the usefulness of FIA data and estimates. Forestry professionals have developed a comfort level associated with using precision metrics when evaluating inventory results for making decisions, making their inclusion with FIA estimates critical. The primary factors that affect precision, confidence interval widths and, therefore, the efficiency of the estimation, are the sampling design, the sample size, the statistical estimator, and the mode of inference of which we consider two, design-based inference, and model-based inference [33].

*1.5. Design-Based Inference*

The key features of design-based inference are that each population unit is assumed to only one possible observation, and inferential validity derives from a probability sampling design as the source of randomization. Further, design-based estimators are generally unbiased or nearly unbiased, meaning that the mean of estimates over all possible samples obtained using the same sampling design and sample size equals the true value [33]. However, even with an unbiased estimator, the estimate for any particular sample may still deviate substantially from the true value. NFIs typically use one or more of three design-based statistical estimators: Simple expansion estimators, post-stratified estimators, and model-assisted estimators [34].

1.5.1. Simple Expansion Estimators

The simple expansion estimators are the most familiar, are used with simple random and systematic samples, and incorporate no RS or other auxiliary data. These estimators are often used to illustrate the relationship between inventory design and statistical efficiency via a fundamental precision metric, the variance of the mean. For a simple random sample, the simple expansion estimator of the mean is expressed as

$$\hat{\mu} = \frac{\sum_i^n y_i}{n} \tag{1}$$

where $y_i$ is the observation of the attribute of interest on plot $i$, typically expressed on a per unit area (hectare) basis, and $n$ is the sample size. The variance estimator for the estimate of the mean is expressed as

$$\hat{v}(\hat{\mu}) = \frac{s^2}{n} = \frac{\sum_i^n (y_i - \hat{\mu})^2}{n(n-1)} \tag{2}$$

where $s^2$ is the sample variance. The half width of the confidence interval ($CI_{hw}$) of the mean, which is based on the variance of the mean, is a commonly used precision index, and is defined as

$$CI_{hw} = t_{1-\frac{\alpha}{2}, n-1} \sqrt{\hat{v}(\hat{\mu})} \tag{3}$$

where $t_{1-\frac{\alpha}{2}, n-1}$ is the 100·(1− $\frac{\alpha}{2}$)th percentile of the $t$ distribution with $n$−1 degrees of freedom. Using survey sampling theory, a user evaluating an estimate of a mean obtained from an NFI in the context of its confidence interval knows that, had the survey been performed a very large number of times with identical methodology, 100·(1−$\alpha$)% of the confidence intervals generated would contain the true, albeit unknown, value of the population mean [35]. For example, FIA's state reports typically report uncertainty in terms of relative standard errors, which are approximately equivalent to 68% CIs. Uncertainty reporting helps the user make informed decisions by taking into account uncertainty in the estimate caused by some combination of the sampling protocol (sample size or plot design characteristics) and variability of the attribute in the population.

1.5.2. Post-Stratified Estimators

If the intent is to increase the precision of estimates and, thereby, reduce confidence interval widths, additional information must be acquired. Multiple approaches for acquiring and using additional information are possible. As per Equation (2), variances can be decreased and precision can be increased by increasing the sample size, $n$. However, doing so incurs additional costs. Preferable options are to acquire the additional information with minimal additional cost. One option is to revise the plot configuration so that each plot captures more landscape information. This option typically entails increasing the plot area or using a cluster design wherein plots are divided into spatially separated subplots with the same aggregated area as a large single plot. However, depending on the spatial correlation of the attribute of interest, the additional information acquired may be less proportionally than the additional area inventoried.

Another option is to use auxiliary information in the form of strata that are related to the attribute of interest. Stratified random sampling can then be used to optimize the spatial allocation of plots relative to a criterion such as variance. The stratified estimator of the mean takes the form

$$\hat{\mu} = \sum_{h=1}^{H} w_h \hat{\mu}_h \tag{4}$$

where $h$ indexes the strata, $w_h$ is a weight proportional to the area of the stratum, and $\hat{\mu}_h$ is the within-stratum mean. The variance estimator is

$$\hat{v}(\hat{\mu}) = \sum_{h=1}^{H} w_h^2 \frac{\hat{\sigma}_h^2}{n_h} \tag{5}$$

where $n_h$ is the within-stratum sample size and $\hat{\sigma}_h^2$ is the within-stratum variance of observations around the stratum mean.

Because most NFIs use at least a large proportion of permanent plots and have a long history of measurements for those same plots, they are reluctant to interrupt or lose those histories. As a result, NFIs seldom use stratified sampling designs because stratifications based on landscape or land cover features change over time and would require re-allocation of plots to strata with each new inventory cycle, thereby producing a loss of historical continuity for at least some plots.

However, for simple random and systematic sampling designs, a large proportion of the benefits of the additional information in strata can still be realized by assigning plots to strata independently of or subsequent to the sampling and then using the stratified estimator in Equation (4) and a variance estimator similar to Equation (5). This technique, characterized as post-stratification, has a long FIA history beginning with double sampling for post-stratification using RS data in the form of aerial photography as the source of stratification data in the 1980s [36], and more recently using satellite data as the source of stratification data [37,38]. FIA currently uses post-stratified estimation as its standard tool for calculating estimates of status of and trends in forest attributes [39]. The advantage of stratified approaches is that they use the additional auxiliary information to increase precision, but a disadvantage is that the rather coarse level at which the auxiliary information is aggregated does not fully exploit its potential.

### 1.5.3. Model-Assisted Estimators

The model-assisted regression estimators use the additional auxiliary information at the plot level, which represents a finer scale of aggregation than do strata. With these estimators, the attribute of interest is predicted for each plot using a parametric prediction technique such as linear or nonlinear regression, a non-parametric technique such as k-nearest neighbors, or a machine learning technique such as random forests. The model-assisted estimator of the mean has the form

$$\hat{\mu} = \frac{1}{N} \sum_{i=1}^{N} \hat{y}_i + \frac{1}{n} \sum_{i=1}^{n} (y_i - \hat{y}_i) \tag{6}$$

where $N$ is the population size, n is the sample size, and $\hat{y}_i$ is the prediction for the $i$th plot. The unbiasedness or near unbiasedness of the model-assisted regression estimator derives from the second term of Equation (6), which compensates for prediction error. Model-assisted variance estimators are based on deviations between observations and their predictions and take the form

$$\hat{v}(\hat{\mu}) = \frac{1}{n(n-1)} \sum_{i=1}^{n} (\varepsilon_i - \bar{\varepsilon})^2 \tag{7}$$

where $\varepsilon_i = y_i - \hat{y}_i$ and $\overline{\varepsilon} = \frac{1}{n} \sum\limits_{i=1}^{n} \varepsilon_i$. If the predictions are sufficiently accurate, deviations between observations and their predictions should be smaller than deviations between observations and strata means as is the case with the stratified and post-stratified estimators or between observations and overall means as is the case with the simple expansion estimators. The result is that model-assisted regression variances and associated confidence interval widths are often smaller than corresponding variance and confidence interval widths for stratified and simple expansion estimators.

McRoberts et al. [40–42] demonstrated the superiority of model-assisted inference and post-stratification for estimates of FIA attributes compared to methods not using RS observations. Magnussen et al. [43], in their own work and a review of several other similar studies, also provide strong support for the use of RS data in a model-assisted context as a way to improve statistical efficiency. The specific mathematical linkages between post-stratified and regression estimation can be seen in Bethlehem and Keller [44], Breidt and Opsomer [45], and Stehman [46]. In addition, McConville et al. [11] provide a tutorial on model-assisted inference for forest inventory that presents post-stratification as a special case of a generalized regression estimator.

*1.6. Model-Based Inference*

The assumptions underlying model-based inference differ considerably from those for design-based inference [47]. First, each population unit has an entire distribution of possible values for an attribute, unlike design-based inference for which there is only a single value. Randomization with model-based inference is realized in the particular value that is observed for each population unit, not via selection of the sample. Finally, the validity of model-based inference is not based on probability samples but rather on the model of the relationship between the attribute of interest and the auxiliary information. An important consequence is that model-based inference can use, but does not require, probability samples. Therefore, model-based inference can be used for small areas for which probability samples are too small for reliable design-based inference and for remote and inaccessible areas for which no probability samples are possible [48]. However, the price to be paid for this greater applicability is that the model-based estimator of the mean is not necessarily unbiased. McRoberts et al. [49] provide the model-based estimator of the mean and estimator of the corresponding variance.

Naturally, the field of model-based inference is very large, and models may take many different forms and operate at different scales of resolution. As an example, there is a growing need for FIA estimates and information over smaller geographic areas, for shorter time periods, and for specific thematic groups. Efforts are underway to expand FIA's capacity to produce estimates for these smaller domains where there are too few plots to use design-based estimators that rely only on data within the domains of interest. Small area estimation (SAE) has been studied extensively in the statistical literature [50]. It relies on indirect methods that borrow strength from outside the domains of interest, integrating both plots and auxiliary RS data through models. There have been numerous examples in U.S. forest inventory history of using SAE methods to essentially increase the effective sample size within small domains. In particular, Moisen et al. [51] applied empirical best linear unbiased predictors (EBLUPs) to estimate forest area and biomass within burned perimeters in the Interior West. Lemay and Temesgen [52] compared imputation methods for modeling basal area and stem density. Goerndt et al. [53] compared synthetic, composite, EBLUP, and most similar neighbor approaches for estimating tree density, diameter, basal area, height, and cubic stem volume using lidar in Oregon. Mauro et al. [54] compared unit and area-level EBLUPs for constructing small area estimates of stand density, volume, basal area, quadratic mean diameter, and height in a western coastal area. Goerndt et al. [55] applied composite estimators to estimate values of FIA attributes relevant to bioenergy production over parts of a 20-state region in the northern US. These are just a few examples, and the field is growing rapidly.

Advantages of model-based inference include its ability to produce defensible estimates for smaller domains and the lack of reliance on probability samples. Disadvantages include its dependence on the assumed model and the consequent potential for bias [33]. Furthermore, those familiar with estimates based on traditional survey samples might prefer them over less familiar uncertainty metrics.

### 1.7. Hybrid Inference

Hybrid inference combines features of both design-based and model-based inference [56,57]. Hybrid inference is used when probability samples are available, but data for the sample units are model predictions with non-negligible uncertainty rather than observations subject to at most negligible measurement error. Hybrid inference incorporates uncertainty from both sources: The model-based prediction uncertainty for the individual sample units and the design-based sampling variability [58,59]. Hybrid inference is more applicable than is generally recognized. For example, FIA plot-level volume data are generally considered "observations" when, in fact, they are really aggregations of tree-level allometric model predictions. Incorporating the effects of this allometric model prediction uncertainty into the uncertainty of large area estimates of mean volume requires techniques such as hybrid inference [58]. Saarela et al. [60] and Ståhl et al. [59] document the utility of hybrid inference for inventory problems for which the model predictions are based on RS data.

### 1.8. Improvement of Economic Efficiency

There are two ways to look at how improving statistical efficiency through RS data integration could improve economic efficiency of the FIA survey. The first is by allowing FIA to meet NFI precision requirements with fewer field plots. For example, the precision targets for the FIA attributes *total timberland area and total volume of timberland growing stock trees* in the Eastern United States are 3% sampling error per 404,686 ha of timberland and 5% sampling error per 28,316,847 m$^3$ of wood volume, respectively [61]. The FIA sample was designed with these targets in mind, but through integration of RS as described in the discussion surrounding Equations (1)–(7), targets could be met in principle with fewer plots. McRoberts and Tomppo [5] give an example of this principle from FIA; they found that in Minnesota, USA, integrating RS data through stratification led to meaningful cost reductions compared to what would have been achieved using a simple random sample. Brooks et al. [62] found that post-stratification for change and forest status variables increased precision by over 100% compared to estimation without using RS data. In the states under the purview of the USFS Rocky Mountain Research Station, it was estimated that there would be an average saving of $300,000 (USD) per state per inventory cycle by transitioning from acquisition and manual interpretation of paper photographs to using RS imagery to perform stratification [63]. The same types of improvements can be shown to occur as the correlation between the attribute of interest and the RS data source increase through improvements in technology. For example, Köhl et al. [64] found that in a study in which a simulated lidar covariate was used to construct regression models for aboveground biomass, more than 800% more plots would be required to achieve 10% sampling error using a model with a coefficient of determination ($R^2$) of 0.3 compared to using one with an $R^2$ of 0.9.

The number of plots needed to meet precision requirements depends on the variable of interest, with attributes with highly variable values requiring more plots. Therefore, for the same number of plots, different attributes will have different precision estimates. A decision to reduce the number of field plots collected is therefore complicated, and entails costs associated with administrative overhead, loss of historical continuity, and disappointment of stakeholders who are using the data for a multitude of purposes including calibrating or validating RS-based models. However, in the event of budget reductions, RS integration and resulting precision improvement may help offset the negative impacts of forced cuts in field work. Perhaps more importantly, RS can add value by exceeding the precision targets that were initially set for the key attributes, and improving precision estimates for all other attributes that, without RS integration, would not have been as useful for making decisions.

The second economic benefit of RS data integration is also related to adding value to the program, through investment in products that go beyond traditional inventory summaries and analyses. Cost savings from reductions in required sample size can be invested in the production of geospatial products such as wall-to-wall maps and models that are used in myriad ways, examples of which are described above [9]. Maps are now part of the currency used for making decisions—they not only provide a synoptic depiction of the resource being reported, but they also provide tools that can be used by others through Geographic Information System (GIS) analyses. It is the pursuit of these efficiencies (statistical and economic) that have led to many of FIA's technological advances in RS, and will lead to further innovation.

A third benefit of RS data integration comes in the form of avoiding unnecessary field work. For example, FIA already makes substantive use of aerial photographs to collect observations on nonforested locations, reducing substantial travel costs to places where no physical measurements are required. Furthermore, RS offers promise for measuring forest attribute data that lend themselves to remote observation, such as land use and land cover change. Examples of these types of processes are given in the following sections.

It should be noted that integrating RS data also can incur costs. If imagery needs to be purchased (as was the case with Landsat satellite imagery in the past), it is especially costly. New computer software and hardware, additional computing requirements, and capacity building may also be required to implement the RS integration. However, in modern times, data sharing policies often result in imagery at no cost to the user, and sufficiently-powerful and ubiquitous software and hardware make RS integration costs relatively small.

## 2. Progression of FIA's Use of RS Data: Inception to Modern Times

### 2.1. Early Use of RS Data in FIA

FIA and other inventory programs have used RS observations operationally for decades. As RS technology has evolved, technologies to take advantage of new, higher quality, and more voluminous data have emerged. In this section, we first discuss the use of photointerpretation from analog sketch maps to aerial photography. Digital imagery from a variety of satellites allowed for improvements; we focus the discussion in this section on research activities and operational products from AVHRR, Landsat, and MODIS.

### 2.2. Photointerpretation (PI)

The earliest maps of forests in the United States were manually drawn from a combination of field reconnaissance and a primitive form of ocular RS by early census takers and cartographers in the late 1800s [65]. After World War 2, however, aerial photography became widespread and its use in forestry expanded greatly [66]. Although it was used for field work planning and other logistics, its principle use in FIA was as the first phase of what is termed two phase or double sampling [34,67,68]. In this statistical method, a set of points located on a dense grid were interpreted and assigned an attribute class, such as a land use or tree volume class. These classes were aggregated into strata that were useful for allocating plots to the landscape in ways that were more economical than would occur without using the technique. Most of the state-level inventories associated with what was to become the FIA program were performed using variations of this principle [69].

A noteworthy, value-added analysis produced with aerial imagery was conducted in the Northeastern FIA region by Riemann and Tillman [70]. In this unique study, PI points were assigned values for forest fragmentation metrics, and the fragmentation status of several Northeastern states was characterized. In the late 1990's, high resolution digital aerial orthophotos from the USGS became widely available for download or ordering on compact discs (CDs) or digital versatile discs (DVDs) [71]; this innovation made FIA pre-field logistical work and research studies that rely on aerial imagery much easier by allowing for analyses such as double sampling or PI of fragmentation to be conducted

in a GIS. In the early 2000's, the USDA National Agriculture Imagery Program (NAIP) began to produce nationwide, high resolution digital aerial imagery on a recurring basis, opening up many new research and applications opportunities for FIA [72].

## 2.3. AVHRR

The cost of PI was large; in order for FIA's double sampling estimation approach to work, hundreds of thousands of points needed to be interpreted on thousands of paper photographs by large teams of full time interpreters. Peterson et al. [63], Reams and Van Deusen [73], and Wynne et al. [74] discussed the benefits of transitioning away from aerial photography toward operational use of digital satellite imagery in forest inventory; these included cost improvements, avoidance of reliance on national aerial imagery programs, and the superior temporal, spatial and spectral coverage of satellite RS data. However, Teuber [75] identified a major challenge associated with the operational use of RS imagery over large areas like those covered by FIA: data volume. To mitigate this concern, Teuber [75] proposed large area mapping with 1 km pixel data from the Advanced Very High Resolution Radiometer (AVHRR), which was carried on National Oceanic and Atmospheric Administration (NOAA) polar-orbiting weather satellites [76,77]. Software and hardware tools to process AVHRR data had been extensively used during the 1980s to make regional maps of forest attributes [78], with the goal of operationalizing the procedures. Teuber [75] found that AVHRR-derived forest area estimates were similar to those from the FIA program for several southeastern states, a promising finding that gave momentum to efforts to move toward implementing satellite RS data usage in FIA.

Research began to show how FIA could operationalize the use of AVHRR. For example, Zhu [79] found that model-based estimates of forest area from AVHRR data compared well with traditional FIA-based estimates for parts of the southern U.S. Roesch et al. [80] successfully used model-assisted inference with AVHRR imagery and FIA data to update county-level estimates of forest area between inventory cycles in Alabama. Moisen and Edwards [81] found that using AVHRR-based maps for stratifying FIA plots to improve estimates was only marginally less efficient than using more costly methods. Several studies employing model-based mapping with FIA and AVHRR data ensued, most notably, that of Zhu and Evans [82], which combined FIA and AVHRR data to produce the first 1 km pixel, nation-wide map of forest type. Cooke [83] built upon the Zhu and Evans [82] approach to refine the AVHRR-based models for parts of Texas and Oklahoma, compared results to FIA estimates, and proposed a forest mapping system that could be operationalized for use in FIA.

## 2.4. Landsat

Data from the Landsat satellites became more heavily used by FIA during the 1980s and 1990s, as well. Landsat 4, carrying the Landsat Thematic Mapper (TM) sensor, was launched in 1982, offering the 30 m × 30 m pixel size spatial resolution that aligns more closely with phenomena of interest to foresters than did most alternatives [84]. There were two problems with operational use of Landsat in FIA: cost and processing capabilities. In terms of cost, The Landsat Remote Sensing Commercialization Act of 1984 led to the privatization of Landsat data, and thus costs became an impediment, as scenes could cost hundreds of dollars each. Lister [85] describes a typical workflow for going from raw Landsat scenes to a finished large area map of forest cover; this includes obtaining multiple scenes from the same time period to deal with cloud cover, and from different time periods to exploit phenological information. Even with government-wide cost sharing agreements, the cost to purchase this large quantity of scenes impeded widespread adoption of the technology beyond small study areas for many government users, including FIA [84].

None-the-less, several state- and region-scale projects using Landsat data were conducted by FIA or partners. Cooke and Hartsell [86], in preparation for transitioning to the use of Landsat data for FIA estimates of forest area in Georgia, explored the relationships between FIA plot geometry and spectral properties of the Landsat data. McRoberts et al. [37,87,88] described an approach to using classified Landsat imagery for improving FIA estimates through stratification. A series of

tests based on assigning FIA plots, under a post-stratified design, to strata constructed with Landsat data were performed around the country by, for example, Hansen and Wendt [36] for the states of Indiana and Illinois, Hoppus et al. [89] for the state of Connecticut, Moisen et al. [90] for parts of FIA's Intermountain West region, Dunham et al. [91] in the Pacific Northwest, and Cooke and Jacobs [92] in Georgia. These tests showed that it was feasible to operationalize the use of Landsat and other RS data-based maps in a variety of forest ecosystems to improve precision via stratification.

Other mapping agencies saw the value of partnering with FIA to help calibrate and validate their Landsat-based maps, and FIA saw the benefits of leveraging these partnerships to obtain maps that met its needs. For example, the Multi-Resolution Land Characterization (MRLC) Consortium was tasked with producing a National Land Cover Database (NLCD) consisting of maps and other products that characterize the nation's land cover [93,94]. For the 2001 NLCD products, MRLC prepared Landsat and other data, and FIA provided plot data for both calibration and validation of maps [95]. While FIA scientists had previously demonstrated their ability to produce their own maps, it became evident that leveraging NLCD maps, and partnering with the MRLC so that FIA needs were more likely to be met, was more efficient economically. The knowledge base associated with using NLCD in FIA workflows grew to the point where NLCD maps were adopted as the foundational layer for FIA's use of post-stratification for most of the country during the first decades of the 2000's [37,69,89,91,96].

Other, later examples of how FIA partners combined FIA data with Landsat imagery for national scale mapping include the multi-agency Monitoring Trends in Burn Severity (MTBS) project [97] and the Landscape Fire and Resource Management Planning Tools (LANDFIRE) program [98,99]. The goal of MTBS is to map fire severity and extent of fires in the U.S. since 1984, using information derived from Landsat imagery. LANDFIRE, which completed its first national product in 2009 and uses MTBS data, produces national-scale geospatial data for use in fire planning, management, and operations. LANDFIRE is an invaluable resource to the fire community, and is heavily based on FIA data for calibration and validation [100,101]. FIA scientists have likewise used LANDFIRE and MTBS products in their research. For example, Whittier and Gray [102] and Shaw et al. [103] intersected FIA plot data with MTBS maps in order to characterize the effects of fire on forests. Dugan et al. [27] used FIA data, Landsat imagery, MTBS, and other data to characterize the disturbances associated with carbon loss on USFS NFS land. All of these applications provide evidence for the value of partnerships with agencies outside of FIA to both acquire expertise and gain access to data products that would otherwise be unavailable, and of the value of using FIA field plot data with RS data.

## 2.5. MODIS

The 1999 launch of the Moderate Resolution Imaging Spectroradiometer (MODIS) on NASA's Terra satellite spurred further innovation into RS integration with FIA. MODIS imagery had several advantages over Landsat data at that time: It was free, it had a near-daily revisit cycle, and it had large swath widths. However, its disadvantage was its spatial resolution—the pixel sizes in wavelengths relevant to vegetation monitoring are 250 and 500 m [104], which, though better than AVHRR, are still quite large. None-the-less, trials of synergistic uses of MODIS with FIA data were begun in the early 2000s. Initial work focused on comparisons of FIA estimates with those derived from MODIS products. For example, White et al. [105] conducted an accuracy assessment of a 500 m pixel MODIS tree cover product (Vegetation Continuous Fields, or VCF) using over a thousand FIA plots. This study, and that of Nelson et al. [106], which also compared FIA to VCF estimates of forest area, found a linear relationship between FIA-based estimates and those from classified VCF products, but that VCF overestimated tree cover for low values of FIA forest cover and underestimated for large values. This was likely due to the effects of spatial resolution on map-based estimates [107] or the spatial mismatch between FIA plots (which consist of triangular clusters of four 0.017 ha subplots distributed within a 44 m radius circular area) and the MODIS pixels (which, were, for these studies, square areas 500 m on a side) [105].

Interest in issues surrounding geometric misalignment between plots and pixels was particularly relevant for MODIS imagery, as its resolution was much closer to the scale of forest patches and

FIA plots than that of the unwieldy 1 km AVHRR pixels. This observation motivated research by Nelson et al. [107] into relationships between pixel size and correlations with FIA estimates; they found that the finest resolution tested (30 m pixels) led to estimates that aligned closest with those of FIA, whereas, surprisingly, larger pixel sizes (90–150 m) led to better estimates overall. That finding aligns with the recommendations of Hoppus et al. [89], who suggested that a 90–150 m pixel aggregation area is most appropriate for intersection of FIA plots with pixels; it also suggests that, despite the geometric incongruence, it was potentially feasible to use 250 m MODIS products with FIA data to meet business needs.

More research into the use of MODIS data for FIA business processes ensued. For example, Liknes et al. [108] assessed the value of MODIS for post-stratification of FIA plots to improve precision of forest area estimates, and found that MODIS imagery performed similarly to that of Landsat. However, Holden et al. [109] assessed the use of a 250 m pixel MODIS-based biomass map for stratification in the North Central FIA region and found little benefit compared to a simple random sampling approach with no stratification. Goeking and Patterson [110] describe the operational use of MODIS imagery for post-stratification of FIA data in the Rocky Mountain region. It is noteworthy that many of the above-mentioned studies show stronger benefits of RS integration for improving precision of forest area estimates, but smaller precision benefits for estimates of attributes like volume or biomass. FIA has traditionally used one stratification layer for both types of estimates, and further research is needed to determine the value of using different stratification layers for different attribute types.

*2.6. Growth of Machine Learning*

During this period of rapid increase in the availability of data from Landsat, MODIS and other sensors, a concurrent blossoming of computer storage and processing power was underway. These increased capabilities brought software tools from the field of what came to be called machine learning to the forefront in the RS community; model building and application with complex, nonlinear algorithms became accessible to the RS practitioner like never before. Previously, relatively simple classifiers, such as rudimentary supervised (based on multivariable nearest neighbor analysis) and unsupervised (based on automated cluster analysis) classification algorithms were commonly used, mostly because their implementation requirements were met by computing capabilities of the time [85]. The expansion of computer storage and processing power made more complex algorithms accessible to RS scientists, and as interest grew in improving environmental monitoring, scientists and statisticians from other fields were drawn to study RS and began to develop and apply techniques and algorithms such as those from the field of data mining and ecology [111].

FIA began to invest heavily in machine learning during the early 2000's. For example, Moisen and Frescino [112] integrated much of the institutional RS knowledge that FIA had accumulated up to that point in a study that combined FIA plots with AVHRR, TM, and digital topographic data to test the ability of several machine learning algorithms to produce maps of various forest characteristics for several states in the Western US. This research demonstrated the feasibility of applying previously-inaccessible algorithms to FIA data, and again highlighted that even after the turn of the 21st century, one of the main impediments to operationalizing these mapping techniques was computer processing speed.

One way these processing speed limitations were overcome was to use simpler algorithms and leverage existing software. For example, McRoberts et al. [113] implemented a k-nearest neighbor algorithm (k-nn), which is a machine learning technique based on multivariate similarity between pixel values associated with FIA plots and those without plots, to produce maps of FIA attributes. Implementing this technique is relatively straightforward in computer languages well-suited for matrix calculations, or through the adaptation of existing software packages such as Erdas or ArcGIS, as was done by Lister et al. [114] to produce k-nn-derived maps for New Hampshire. Ohmann and Gregory [115] used a k-nn-like approach with FIA data and Landsat to create maps of imputed estimates

of forest attributes in Oregon. Nelson et al. [116] compared the stratification efficacy of forest/nonforest maps derived from NLCD with those from several machine-learning approaches, including k-nn.

New data processing tools and scripts streamlined workflows for combining plots and pixels and applying machine learning models to RS data. This included the development of freeware, such as spatial analysis functions for the R statistical software [117], as well as the adaptation of various commercial packages to work with RS data [118]; several of these were reviewed by Ruefenacht et al. [119]. These advances allowed for the achievement of a milestone for the FIA RS community: the production of the first national-scale map of forest biomass [120] and forest type [121]. The production of these maps demonstrated that it was feasible for FIA scientists and partners, working together, to achieve the production of national-scale maps integrating FIA data and RS observations.

A second major milestone for FIA was achieved by Wilson et al. [122], who produced national-scale maps of forest tree basal area by species using MODIS and other GIS data, and by Wilson et al. [123], in which nationwide maps of carbon stocks were produced. As with the work by Moisen and Frescino [112], these achievements were built using institutional knowledge acquired over the previous decades. The study built upon machine learning work done with FIA plots [113–115], accumulated experience with MODIS [107,120,121], and a series of past workflow development strategies for linking FIA plots, RS imagery, and various software platforms for GIS-based modelling. A similar effort was conducted by the USFS Forest Health Technology Enterprise Team (FHTET) [124]; the FHTET product was used in the production of the National Insect and Disease Risk Map [125], which is used by forest managers and policy makers to understand the potential impacts of forest pests. The role of FIA data in the production of the Blackard et al. [120], Ruefenacht et al. [121], Wilson et al. [122], and Ellenwood et al. [124] maps not only showed what could be done, but showed that these new techniques could be operationalized and made part of a suite of standard, FIA-based products.

*2.7. Advanced Uses of Landsat*

2.7.1. Opening of the Landsat Archive

The most important stimulus of research and development of Landsat-based science was the 2008 change in the U.S. government's Landsat Data Distribution Policy to allow for the release and download of the entire Landsat archive [126,127] at no cost to users. Coupled with enhancements in web-based RS data discovery and bulk ordering and download tools like USGS's Earth Explorer [128], obtaining Landsat data, which had previously been expensive and often irksome, had become nearly effortless. Similar mechanisms were developed by the European Union's Copernicus program [129], and nascent attempts to provide analysis-ready data, such as the Web-Enabled Landsat Data (WELD) [130], facilitated use and research of mosaicked, cloud-corrected, normalized imagery. Along these same lines, the Landsat archive was being pre-processed with various high-level algorithms like the Landsat Ecosystem Disturbance Adaptive Processing System (LEDAPS) to make data easier to use and facilitate retrospective analyses and monitoring studies [131,132].

2.7.2. Vegetation Change Tracker and the North American Forest Dynamics Project

The theoretical ability to obtain the entire Landsat record, pre-processed in useful ways, led to some ground-breaking research into what have become known as Landsat time series (LTS) for forest monitoring [133]. LTS consist of stacks of Landsat scenes, collected at a regular time interval over a long period, allowing for the tracking of the value of some image characteristic like a spectral index for each pixel and scene in the stack [134] (Figure 1). FIA became involved in nationwide use of LTS data for forest monitoring through participation in the North American Forest Dynamics (NAFD) project as a way to reduce spatial and temporal uncertainty in carbon estimates related to forest dynamics [135–138]. FIA data were used throughout the decade-long NAFD project to validate year of disturbance estimates from the LTS-based models [139], to refine spectral thresholds to detect trees in xeric systems [140], to interpret ratios of forest disturbance area to removal volumes across

regions [137], to investigate annualized timber product outputs [141], to annually map the magnitude of harvests using repeat FIA basal area measurements [142], and to help attribute causal process to disturbance events [139,143].

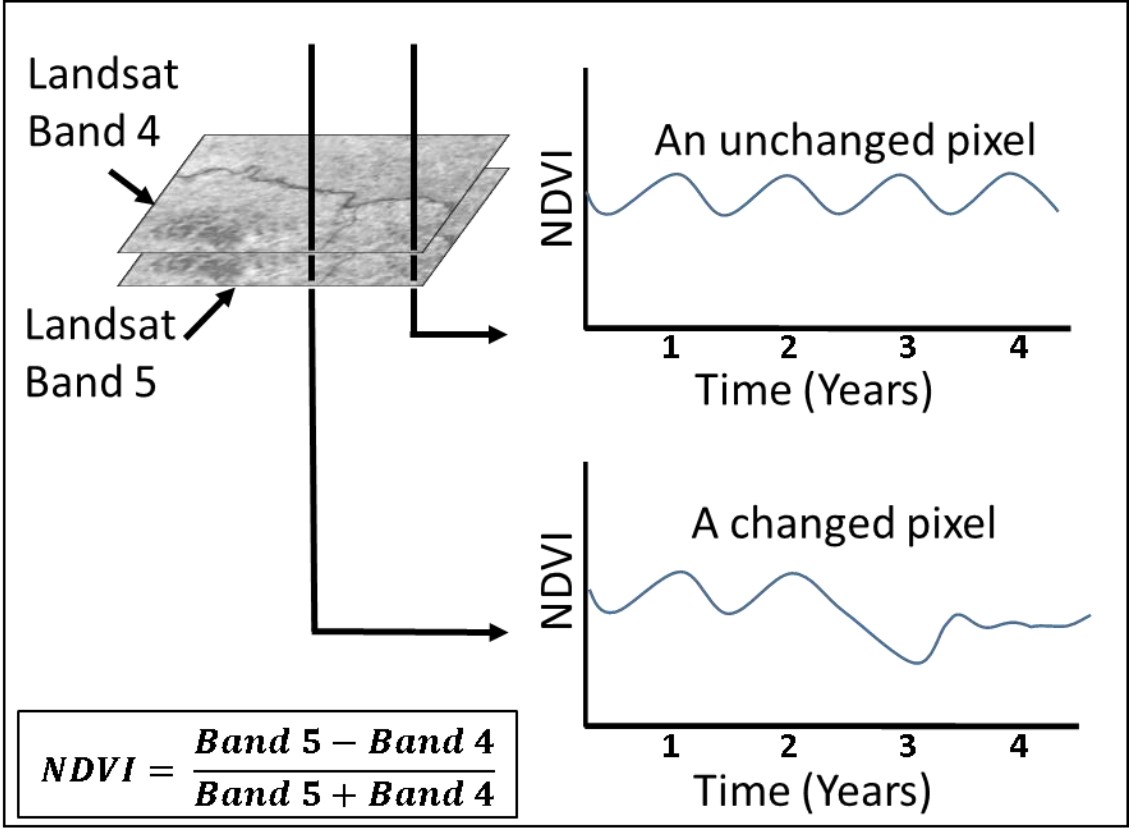

**Figure 1.** Conceptual diagram of the creation of Landsat time series (LTS) analysis inputs. For each pixel, brightness values for a user-specified combination of layers is measured for each scene in the record (for Landsat 8, a new image is acquired every 16 days). Normalized Difference Vegetation Index (NDVI), a commonly-used vegetation index, can be calculated for each scene from Landsat 8's bands 4 and 5, and plotted against time. The properties of the resulting time series can be assessed to identify changes through time.

The Vegetation Change Tracker (VCT) algorithm [134,144], used to generate the NAFD LTS and annual change maps, was the first algorithm with nationwide application to reliably identify moderate through severe forest canopy loss events by detecting statistical anomalies in the LTS spectral signatures. These NAFD products, now archived at the Oakridge National Laboratory [136], allowed FIA projects to capitalize on annual LTS and change maps. For example, Stueve et al. [145], in order to improve annual estimates of forest change, adapted VCT by using information from snow-covered areas in the Lake Superior and Lake Michigan basins. For the same area, Garner et al. [146,147] then used the modified VCT data to improve the NLCD forest cover classes by subdividing them into successional stages, and Tavernia et al. [148] used these modified maps to characterize early successional habitat using FIA data. In a similar study, Nelson et al. [149] used VCT products to identify and characterize young forests in Wisconsin. Powell et al. [150] used NAFD and FIA data to quantify time series of live aboveground forest biomass dynamics. Brown et al. [151] used VCT products to assess impacts of surface coal mining on forest distribution in West Virginia. Schroeder et al. [152] investigated using FIA plot data, time series observations, and annual change maps to improve estimates of forest disturbance using model assisted post-stratified estimation. Moisen et al. [153] modified a statistical technique using constrained spline fitting and temporal shapes building off of NAFD data to model, map and

monitor forest dynamics over three decades. A methodology for forest change process attribution, building off of NAFD LTS and an ensemble of change detection algorithms, was described and piloted in Schroeder et al. [143]. Dugan et al. [27] and Birdsey et al. [28] combined FIA data with VCT and similar outputs to characterize carbon dynamics by disturbance type for U.S. National Forests. In 2020, Schleeweis et al. [139] delivered nationwide maps attributing forest disturbance type.

### 2.7.3. TimeSync and LandTrendr

Two other tools were being developed by the LTS research community around the same time VCT was created: LandTrendr [154] and TimeSync [155]. LandTrendr (Landsat-based detection of Trends in Disturbance and Recovery) uses the then novel principles of exploiting LTS data to model trajectories of change, capitalizing on breaks in the trajectory trends (rather than statistical anomalies), which allow segments and vertices to capture and characterize long slow declines as well as abrupt losses. TimeSync is a visualization and database tool aimed at providing image interpreters with an environment and suite of tools to visually assess multi-band spectral signatures through time and record labels of land cover and land use for individual breaks (vertices) and trends (segments) over decades of imagery (radiometrically calibrating, cloud and shadow filling and chipping images along the way). FIA quickly saw the value of these tools, and the potential benefits of conducting LTS analyses on FIA plots to relate forest attribute data with temporal and spectral signatures from LTS. For example, Ohmann et al. [156] used LandTrendr to radiometrically correct Landsat scenes and to identify forest change as part of an old-growth forest change mapping effort. Schroeder et al. [152] demonstrated how manual interpretation of LTS imagery improved estimates of area and type of forest canopy disturbance in the Uinta Mountains of Utah; they collected a new variable (evidence of past disturbance) from the LTS using an approach modeled on TimeSync, and used maps derived partially from LandTrendr for post-stratification for variance reduction. Bright et al. [157] used LandTrendr to estimate characteristics of forests affected by bark beetle in five western states. Bell et al. [158] used an improved version of TimeSync [159] to study historic drought effects on forest canopy decline. Zhao et al. [140] used FIA data with the TimeSync tool to assess historic forest cover changes and their causes. Schroeder et al. [143] used TimeSync data, FIA plots, and outputs from VCT to classify forest disturbance causality in ten Landsat scenes distributed across the United States. Gray et al. [160] used TimeSync to identify and model historical changes in aboveground woody carbon on FIA plots. Filippelli et al. [161] used LandTrendr data with FIA plots to assess historical trends in pinyon-juniper biomass across parts of several western states. Research into the use of these technologies, as well as into the use of time series data from the Sentinel-2 mission for tracking forest change [162], is continuously evolving.

### 2.7.4. Landscape Change Monitoring System

The Landscape Change Monitoring System (LCMS) project, which grew out of the multi-agency Monitoring Trends in Burn Severity (MTBS) project [97], was a USFS-led initiative to exploit the LTS record by using machine learning algorithms to map landscape change. One of the main premises behind LCMS was that although different image classification algorithms will lead to slightly different maps, each with their own strengths and weaknesses, the ensemble of maps made from these algorithms can be summarized to improve upon any of the individual maps that constitute the ensemble. Schroeder et al. [143] and Cohen et al. [163] showed how using ensemble approaches for change detection improved upon algorithms that used individual algorithms and/or spectral indices; they combined several LandTrendr outputs in order to assess the impacts of various band and spectral index choices on classification accuracy. Schleeweis et al. [139] used an ensemble approach exploiting different algorithms for detection and classification of the type of disturbance processes occurring nationally, based on LTS analysis. Healey et al. [164] used an ensemble approach that leveraged the strengths and weaknesses of multiple algorithms, balancing their omission and commission errors, to create an improved overall model and map of forest change locations through time. Based on this initial work, an operational, cloud-based LCMS Data Explorer web application (http://lcms.forestry.oregonstate.edu/)

was created by the Forest Service. This tool was designed as a way to analyze, subset, and download LCMS data products, which include forest cover loss by year since 1984.

### 2.7.5. Use of LTS-Derived Covariates for Mapping of FIA Attributes

Additional exploration of the fusion of LTS and FIA data for FIA attribute mapping includes work by Wilson et al. [165], who used harmonic regression on LTS and FIA data for the state of Minnesota to map several FIA attributes; they found that the regression coefficients derived from Fourier analysis on LTS data improved mapping accuracy in important ways compared to alternatives. Brooks et al. [62] found that LTS-derived maps performed well as the source of strata for post-stratified estimation of FIA attributes. Derwin et al. [166] used harmonic regression coefficients from LTS in a canopy cover estimation approach, and found that they performed better than other LTS approaches when compared to FIA data. Research into creative uses of LTS for improving FIA business processes is in its early stages.

### *2.8. Cloud Computing*

### 2.8.1. Cloud-Based Data Processing

One of the most significant advances in RS data analysis over the last several decades is the development of a cloud-based image processing and storage system called Google Earth Engine (GEE) [167]. GEE takes advantage of high-performance, parallel computing systems and the petabytes of RS data from several archives to create a cloud-computing environment for image storage and analysis. Previous workflows for large area mapping involved ordering, downloading (or mail ordering CDs or DVDs), pre-processing, mosaicking, and applying algorithms to often dozens of images on personal computers or local servers [85] (Figure 2). GEE removes several of these steps by hosting analysis-ready, mosaicked RS imagery data on distributed servers, and providing access to tools for pre-and post-processing, building models, and applying the models to the imagery. This has made large area mapping on a repeated basis accessible to a broad user community. Before FIA actively began using cloud computing, Hansen et al. [168] used GEE for forest mapping, creating yearly, global, LTS-based tree cover change maps between the years of 2000 and 2012 using machine learning algorithms applied to more than 650,000 Landsat scenes (20 terapixels of data processed using one million CPU-core hours). Since then, this approach has been operationalized and incorporated into an annual tree cover monitoring system called Global Forest Watch (GFW) [168,169]. GFW is particularly well-suited to detect change in tree cover in areas with continuous canopy cover.

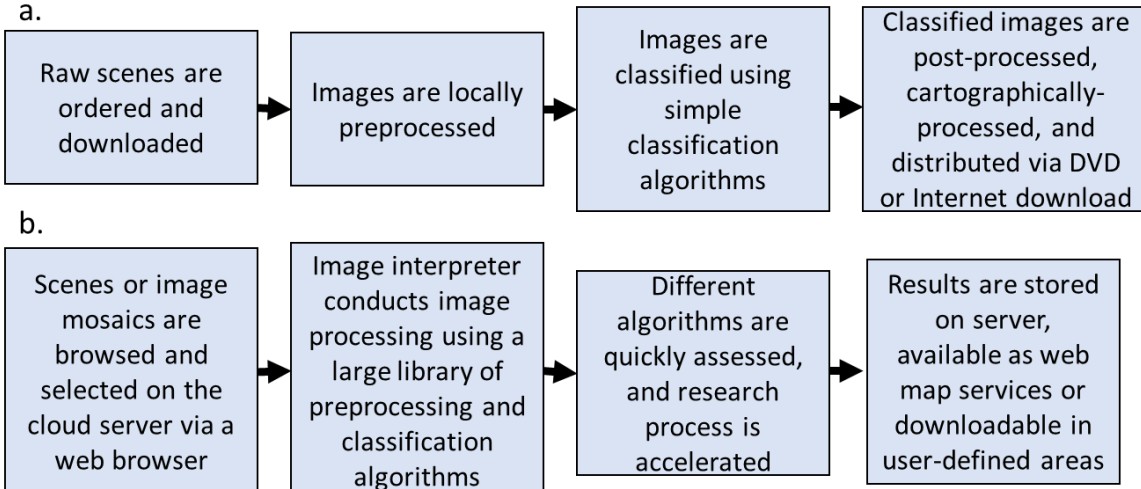

**Figure 2.** Description of two processes for conducting remote sensing. (**a**) Traditional approach, prior to era of cloud computing; (**b**) Workflow that is used during the era of cloud computing.

The trend of operationalizing change detection algorithms with GEE continued with subsequent conversions of TimeSync [155], LandTrendr [170], and LCMS ensemble model [164] code and workflows to GEE code, along with those of another algorithm, Continuous Change Detection and Classification (CCDC) [171]. CCDC uses a two-step cloud, shadow, and snow-masking algorithm, along with various metrics of spectral change from LTS data, to continuously detect change as new scenes are acquired. The use of CCDC algorithms on cloud computing platforms and supercomputers allowed for the full implementation of the U.S. Geological Survey Land Change Monitoring, Assessment and Projection (USGS LCMAP) initiative [172,173]. LCMAP, like the LCMS ensemble-based maps, seeks to provide an operational landscape change monitoring system; the approaches differ based on differences in the algorithms employed, the classification systems, and the fact that LCMS focuses mostly on forests.

### 2.8.2. Cloud-Based Data Hosting and Serving

The NAIP program's transition of CD- or DVD-based digital aerial imagery to online imagery was the harbinger of a paradigm shift toward serving massive datasets to users through web services. The most commonly-used web map services protocols are the Web Map Service (WMS), Web Feature Service (WFS), and Web Coverage Service (WCS) [174]. These types of map services allow client software, like a user's GIS, to interact with either pre-rendered images of the geographic entities, or the data associated with the geographic entities themselves. For example, if a user loads a local GIS data file of FIA plot locations, NAIP imagery from an Internet-based server such as the National Map (https://viewer.nationalmap.gov/services/) can be loaded beneath the plots so an interpreter can examine the imagery associated with each plot location. A WFS could also be used to load a vector-based watershed layer as well, so the user can assign attributes associated with each watershed to each plot. Finally, a WCS can be used to load Landsat imagery, and the pixel values of the imagery could be attached to the FIA plots and used for model building or validation. The map services are designed to optimize streaming of the GIS data to the client computer by returning only what is needed (the area and resolution) based on the context of the client GIS field of view; this minimizes latency and consumption of Internet bandwidth. Popular commercial Internet map services that rely on similar principles include Google (Google Maps and Google Earth) and Microsoft (Bing maps), and there are many open-source providers of map services [174] as well.

### 2.9. Increased Use of NAIP

### 2.9.1. Image-Based Change Estimation (ICE) and Logistical Planning Prior to Fieldwork (Pre-Field)

Map service technology allowed for several innovations within FIA and the broader Forest Service, particularly through the enhanced use of the NAIP imagery, which is often served through a WMS [72]. NAIP imagery generally consists of frequently-reacquired, nationwide, 3 or 4 band digital imagery with pixels with 1 m resolution [72]. Using NAIP, pre-field preparatory work became streamlined, with field crews being able to easily access recent imagery of plots online to determine if field visits are necessary and to obtain navigation and context information [175].

Efficient, rapid PI of high-resolution imagery in a GIS also became feasible. Toney et al. [176] compared results from PI vs. field-derived canopy cover estimates and found that the PI estimates were 10–20% larger than those in the field. Rapid PI was exploited by Lister et al. [177] for a trees outside forest (TOF) inventory for the states of North Dakota, South Dakota, Kansas and Nebraska, in which tens of thousands of photo plots were interpreted with respect to the presence or absence of TOF, and the plot labels were used to conduct prestratification to improve the efficiency of a ground survey. In a double sampling for post-stratification context, Westfall et al. [178] demonstrated the value of PI of WMS-based NAIP imagery for improving inventory efficiency in three counties of Pennsylvania. Frescino et al. [179] used digital photographs to conduct an inventory of TOF in Nevada. Lister et al. [180] used a rapid PI approach to monitor land use change in Maryland, and Lister et al. [181] used a similar method to characterize forest degradation in Maryland and Pennsylvania. Nowak and Greenfield [182–184]

characterized land cover and urban tree dynamics at the national scale using rapid PI in Google Earth, and Nowak et al. [185] developed a tool called i-Tree canopy that allows users to create PI projects online using Google imagery as a backdrop. Based on similar principles, the FIA program developed a GIS tool using WMS-based imagery in order to collect canopy cover information for updates to the NLCD canopy cover project and other FIA business processes [186], and Lister et al. [180] developed a software tool for rapidly interpreting land cover on digital photo plots.

Building upon lessons learned from these and similar research efforts, the FIA program operationalized this technology for forest monitoring with the Image-based Change Estimation (ICE) project [187]. ICE was developed partly as a response to a 2009 resolution by the National Association of State Foresters that encourages FIA to increase the use of RS data for monitoring forest cover dynamics [188]. Furthermore, there was a perception among FIA partners that the FIA cycle length, which ranges between 5 and 10 years, was too great to identify important forest cover changes in a timely way. The ICE project addressed this by interpreting imagery associated with each new NAIP cycle, which is generally 3 years long [72]. Estimates include area by land cover, land use, and change category and agent, and are being used to address 2018 Farm Bill objectives of more timely production of estimates of forest trends and increased use of RS technology in monitoring.

### 2.9.2. Pixel-Based Mapping Using NAIP

Automated image classification of high resolution satellite imagery has been of interest in FIA for decades, but issues surrounding its large cost have limited its use. NAIP, on the other hand, is supplied by the USDA at no cost to users, and is therefore an appealing option [189]. Various attempts have been made to combine NAIP imagery with FIA plot information for pixel-based mapping. For example, Meneguzzo et al. [190] compared pixel-based and other classification methods using NAIP, and found that pixel-based classification performed similarly to an ocular photointerpretation method in Minnesota. Hogland et al. [191] used FIA plots as training data for classification of NAIP using a machine learning algorithm, and found that using the NAIP-based map improved estimates of various FIA attributes compared to using plots alone. Hogland et al. [192] used NAIP and its derivatives with FIA data to tree density and basal area for portions of Georgia, Alabama and Florida, and cited the need to download NAIP imagery for local processing as being one time-consuming element of the project. To circumvent this challenge, Chang et al. [193] streamlined pre-processing and acquisition of NAIP imagery by using the cloud-based GEE platform in a project that relied on machine learning to map several FIA attributes in California and Nevada.

### 2.9.3. Object-Based Image Analysis Using NAIP

In addition to manual PI, FIA has used NAIP imagery for semi-automated land cover class mapping using object-based image analysis (OBIA) [194,195] (Figure 3). OBIA works by using algorithms that assign each pixel in an image to a spatially-contiguous cluster, typically a polygonal area that meets a homogeneity criterion used as a parameter in the algorithm. This in effect creates a map of land cover polygons that can then be classified using machine learning approaches. Frescino et al. [189] compared OBIA-based maps produced using NAIP and FIA data with those produced with other high resolution imagery and found that it performed nearly as well as the more costly imagery. Lister et al. [196] fused Landsat and NAIP imagery and performed OBIA to impute forest inventory information to stands in a forest in Maryland. Liknes et al. [197] used NAIP OBIA-based machine learning algorithms to classify tree cover in agricultural areas in North Dakota. Riemann et al. [198,199] assessed the value of using OBIA-based canopy detection products, which partly-relied on NAIP, for estimating canopy cover in several states. Meneguzzo et al. [190] and Meneguzzo [200] found that image segmentation compared favorably with or improved upon pixel-based classification of NAIP imagery to map trees outside of forest. Liknes et al. [201] used a NAIP OBIA-derived map of tree cover patches and a patch shape detection algorithm to identify windbreaks in Nebraska, and Paull et al. [202,203] and Kellerman et al. [204], produced statewide, NAIP-based maps of tree canopy for both Kansas

and Nebraska. These studies show the potential for using the spectral information contained in the multi-band NAIP imagery, but do not fully exploit the structural information that could be obtained from stereo photogrammetric analysis of overlapping NAIP images.

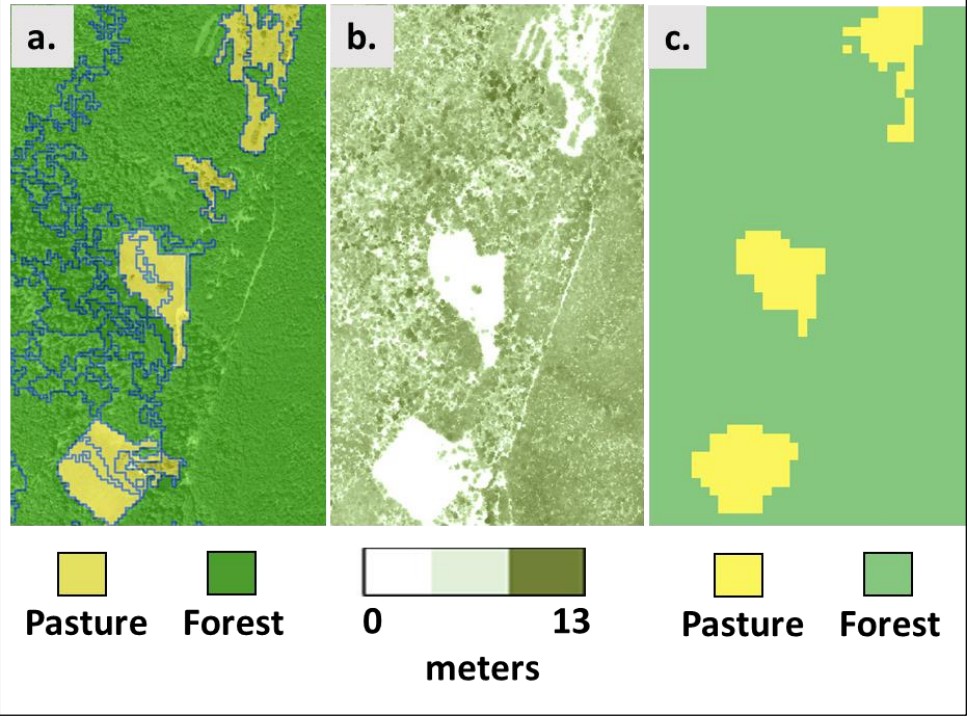

**Figure 3.** Comparison of different uses of National Agriculture Imagery Program (NAIP) for forest resource measurement and mapping with Landsat-based classification. (**a**) A NAIP image with an image segmentation applied (blue polygonal areas) and an object-based image analysis (OBIA) classification algorithm applied to the polygons (yellow is nonforest, green is forest); (**b**) a pixel-based tree height classification of NAIP imagery; (**c**) A Landsat-based forest/nonforest classification.

### 2.9.4. 3-D Processing of NAIP for Structure

New technology in software and processing allows for the development of optical 3-d models from stereo NAIP imagery [205]. The approach works by using advanced software to process digital aerial photography of the same location taken from different angles to create 3-d point clouds that represent the height of objects (such as tree canopies). The advantage of this approach over active RS technology like Light Detection and Ranging (lidar, described in the next section) is that it can be over an order of magnitude smaller in cost [206]. A disadvantage over lidar is that the quality of the information from a NAIP-derived point cloud is typically lower. FIA therefore performed several tests to determine whether derivatives of 3-d NAIP imagery are of value to enhance forest inventory. For example, Gatziolis [207] obtained promising results when using texture metrics from digital aerial photography for estimating canopy structure parameters in a range of forest types in Michigan. Webb et al. [205] investigated the value of NAIP point clouds compared to those from lidar for vegetation mapping and found that they were comparable in certain conditions, but that NAIP in general was not as accurate as lidar. Strunk et al. [206] created NAIP-based canopy height models under different processing scenarios and determined that using the height maps in FIA estimation process for stratification led to a 400% increase in precision of volume estimates over models without RS-based strata.

*2.10. Airborne Light Detection and Ranging (Lidar)*

Lidar methods have revolutionized several aspects of forest inventory over the past several decades. In a lidar system, a laser, typically mounted on an airplane, emits a beam of light, the transmission time between emission of the light, reflection of the light from an object of interest, and its return to the receiving sensor is recorded, and the time data are converted to information about the structural characteristics of the object that reflected the light [208]. There are various types of lidar data types, and they have been used with FIA data for wall-to-wall mapping, sample-based estimation, and field measurement of forest characteristics.

2.10.1. Airborne Lidar for Wall-to-Wall Mapping

FIA data have been used in concert with airborne lidar in various ways for many years. Relationships between the two data sources have been explored by Gatziolis [209,210] and Schrader-Patton [211], who demonstrated that lidar can be used to refine the locations of FIA plots by matching individual tree locations. Gatziolis [212] used lidar to create maps of Site Index (an index related to tree growth potential) and later [210] compared canopy cover estimates from an FIA PI approach with those from lidar and found systematic overestimation for small canopy cover values and underestimation for large canopy cover values. Riemann et al. [198,199] assessed the utility of lidar for replacing PI as a way to estimate canopy cover, and found that there were important differences between PI, field-mapped, and OBIA-based, lidar-dependent estimates, particularly in areas with small amounts of canopy cover. Andersen et al. [213] evaluated the error budgets in lidar-derived estimates of tree height for deciduous species and conifers, while Li [214] and Gopalakrishnan et al. [215] found that the correlation between lidar-derived tree heights and those measured by FIA was high. However, Gatziolis et al. [216] documented discrepancies between lidar-derived and field inventory estimates of tree height in challenging U.S. Pacific Northwest conditions. Possible reasons for the discrepancies found in these studies include field-lidar georeferencing mismatch, variable lidar quality, and field crew method inconsistency.

Due to the correlation of lidar-derived information with forestry attributes of interest, lidar data have often been used as a covariate in predictive modeling studies. The structural information contained in the lidar data can help inform models of forest volume or biomass, canopy cover, and land cover class. One approach to this is to use wall-to-wall maps of lidar as the main input to models. For example, Skowronski et al. [217] used FIA data and lidar in New Jersey to map forest structure and fire fuel loads. Johnson et al. [218,219] used lidar in concert with FIA plots to make high-resolution maps of forest carbon in Maryland. Sheridan et al. [220] summarized lidar data on and in the vicinity of FIA plots and modeled and mapped tree volume and biomass in Oregon. For a study site in Hawaii, Hughes et al. [221] conducted the first study to map aboveground carbon across a tropical landscape with lidar and FIA information. Joyce et al. [222] installed and measured FIA-like plots for producing lidar-based models of coarse woody debris in Minnesota.

Studies over large areas, such as U.S. states, where lidar data are the sole predictor used in models of FIA attributes are relatively rare because lidar data are massive, and significant pre-processing is required to generate an analysis-ready dataset for spatial modelling over these large areas. Airborne lidar data have traditionally been more expensive than optical imagery, and repeat acquisitions supporting assessment of forest dynamics or other forest characteristics that are linked to phenology are infrequent. It is therefore much more common for lidar to be used in concert with other data sources such as Landsat when mapping over large areas. For example, Lefsky et al. [223] combined FIA plot data, lidar, and Landsat to map forest biomass in Oregon and Washington. In a study combining various RS data types and FIA data, Chopping et al. [224] fused MODIS and data from NASA's Multiangle Imaging Spectroradiometer (MISR) to produce maps of biomass and forest structure in the southwestern U.S., and used FIA-based maps and lidar data to assess results. Deo et al. [225] compared LTS, lidar and fused LTS-lidar datasets for back-projecting biomass to a baseline year of 1990, and found that fusion of LTS and lidar improved results over alternative models. In similar studies

that fused lidar, Landsat, and FIA data in four states in the eastern U.S., Deo et al. [226] assessed the advantages of using site-specific models vs. generic models of forest biomass, Deo et al. [227] assessed the impacts of spatial resolution of RS data on predictive accuracy, and Ma et al. [228] compared the performance of models using various combinations of the LTS, lidar, and FIA inputs. Bell et al. [229] used FIA data to compare lidar and LTS maps of aboveground biomass in 3 regions in Oregon and Washington, and found that lidar performed better than LTS. Gopalakrishnan et al. [230] also used LTS and lidar data, but fused them to impute site index estimates to loblolly pine stands over large areas in the southeastern U.S.

### 2.10.2. Airborne Lidar for Sample-Based Estimation

FIA's main source of estimates for reporting is currently not maps of forest attributes (what is colloquially referred to as "pixel counting"), but rather the information from its large network of ground plots. This is due partly to its legislative mandate, historical reliance on traditional survey sampling, and need for consistently reliable long-term data, and partly to the lack of practical ways to quantify systematic error (bias) in ways that are agreed upon by the scientific community [5,33]. However, lidar can also be collected in a sampling mode and be used practically for model-assisted inference, as described above. For example, Andersen et al. [231] used structural measurements from airborne lidar strip sample-based maps made from model-assisted regression as the second stage in a two stage estimation procedure in Alaska. Alonzo et al. [232] demonstrated an application of repeated measurements of airborne lidar samples to assess fire effects over FIA plots in Alaska. Strunk et al. [233,234] demonstrated a regression estimation approach using inventory data and lidar from a study site in Washington. McRoberts et al. [42] employed lidar-assisted regression estimators with FIA data as part of a comparison of different approaches at a study site in Minnesota. In a similar study over the same area, McRoberts et al. [235] assessed the "shelf life" (applicability through time) of lidar data that were not collected contemporaneously with inventory data when conducting model-assisted estimation. All of the applications mentioned in this section so far are based on airborne lidar; however, spaceborne lidar systems can be used in forest inventory applications as well.

### 2.11. Spaceborne Lidar for Sample-Based Estimation

### 2.11.1. GLAS

One of the objectives of the Ice, Cloud, and land Elevation Satellite (ICESat), which carries the Geoscience Laser Altimeter System (GLAS), is to measure vegetation canopy height [236,237]. GLAS data consist of sets of crossing transects that cover much of the Earth, with detailed laser height measurements taken on points distributed along these transects. Lefsky et al. [238] found good agreement between canopy height measurements from GLAS and field-measured heights from modified FIA plots in Tennessee and Oregon, and Pang et al. [239] found strong relationships between canopy height estimates from GLAS, airborne lidar and field measurements on modified FIA plots. Plugmacher et al. [240] found a somewhat weak relationship between FIA and GLAS-based height data in the Appalachian mountains, as did Li et al. [241], who used a GLAS-Landsat fusion approach in young forests in Mississippi. None-the-less, GLAS data have been shown to be useful for measuring not only height, but also forest biomass relative to estimates from FIA data in boreal forest regions of Alaska [242], and, in a unique fusion of FIA and other inventory data, airborne lidar, and GLAS, for the conterminous U.S. and Mexico [243].

### 2.11.2. GEDI

A problem identified by Nelson et al. [243] with the GLAS approach for estimating biomass or other attributes regionally is that calculating confidence intervals is not straightforward due to the combination of models used in the estimation procedure. In anticipation of this challenge, Healey et al. [244] proposed a sampling design that allows for the use of FIA data and nearby GLAS data

for biomass estimation, using the state of California as a pilot study. Other approaches were also developed in anticipation of the GLAS-like data provided by NASA Global Ecosystem Dynamics Investigation (GEDI) mission, which includes a full-waveform lidar instrument that was mounted on the International Space Station (ISS) and was launched in 2018 [245]. For example, Ståhl et al. [59], McRoberts et al. [58], Saarela et al. [246], and Patterson et al. [247] discuss estimators that combine FIA and GEDI data. They use various combinations of wall-to-wall optical data (typically Landsat), a sample of more highly-correlated data (such as GLAS or GEDI), and sparse ground plots (FIA) to produce estimates and confidence intervals that are interpretable through the lens of sampling theory. Similarly, Nelson et al. [243] use a multi-phase sample that combines ground, airborne lidar and spaceborne lidar to generate estimates of biomass that are compatible with those from survey sampling. GEDI and similar datasets like those from the next generation of the ICESat mission (launched in 2018) offer opportunities to enhance or replace certain monitoring activities currently conducted with ground inventory plots, due to the lower overhead of sample-based methods and frequent reacquisition of data.

### 2.12. Unmanned Aerial Systems and Terrestrial Lidar

FIA is currently in the early investigatory phase of research into operational use of Unmanned Aerial Systems (UAS), which consist of aircraft such as drones that carry some combination of active and passive sensor systems for imaging forests at the local scale. For example, Gatziolis et al. [248] used a UAS to develop 3-d models of individual trees and found that photo-based models compared well to more detailed lidar-based 3-d models developed separately. Fankhauser et al. [249] concluded that their use of UAS to measure tree heights and counts showed promise for supporting traditional forest inventories. The largest potential for future use of UAS operationally in FIA is likely their use in remote or inaccessible areas, such as was done by Alonzo et al. [250], who flew drones over FIA plots in Alaska and found that they were moderately effective at estimating forest type, basal area, tree density, and biomass.

Terrestrial lidar is another new technology in its early stages of exploration by FIA. Gatziolis et al. [248] and Strigul et al. [251] compared terrestrial lidar-based tree structure measurements with those from other sensors for testing 3-d tree imaging technologies, and Gatziolis et al. [252] and Klockow et al. [253] used it to gather data related to live and dead tree allometry, respectively. Both found that it offered important improvements over standard FIA methods, and showed potential for improving fieldwork efficiency. Once algorithms are refined and processing software and hardware are adequate, terrestrial lidar and UAS technology will have the potential to improve FIA's plot-based data collection efforts by collecting new structure attributes and eliminating certain manual measurements.

## 3. General Observations on RS Data Integration in FIA and Other NFIs

### General Characteristics of FIA's Use of RS

There are several instructive principles that emerge when examining the evolution of FIA's use of RS technology:

○ **NFI data are invaluable to creating RS products**. They provide a standardized source of training data for models, and their use raises the likelihood that RS-based estimates will align with NFI-based estimates. They also provide valuable validation data for users interested in conducting map accuracy assessments at both the plot-pixel scale, as well as over larger geographic areas like U.S. counties, for which NFI-based estimates and confidence intervals can be generated.

○ **A successful RS program has access to RS data inputs, software, and hardware, including affordable high performance computing systems.** There was a strong correlation between advances in FIA's use of RS and improvements in Internet and personal computer technology, and, more recently, a similar increase in RS technology usage with the opening of the Landsat archive, the advent of other free RS data input sources, and the advent of cloud computing systems. It cannot be understated how the democratization of RS data acquisition and processing

technologies have led to improvements in our ability to monitor forest resources, and how FIA scientists are contributing more and more to both basic and applied research aimed at advancing forest science in these areas.

- **Advances in RS usage require nimbleness and outlets for creative investigation**. Support for intellectual fora such as program meetings and scientific conference attendance advances what McRoberts [254], citing Reichenbach [255], calls the "discovery" component of science, i.e., the exploratory and creative part of the scientific method that focuses on identifying research questions, forming hypotheses, and developing models. Mechanisms for scientists and technical staff to conduct research and share preliminary results in a less-formal way furthers advancements.

- **Advances in RS are incremental, beginning with discovery and leading to operationalization**. Figure 4 is a conceptual model showing the process that FIA RS research has typically gone through over the last several decades, beginning with knowledge discovery and ending in operationalization. It is noteworthy that some of the studies described in this review have not yet, or never will, become operational; Figure 4 identifies several points in the research and development process where operationalization can be impeded:

  (a) After research into methods for application is conducted, it becomes clear that it is not feasible, or results are not as expected due to poorly-conceived research ideas that attempt to integrate components of many studies and stakeholder needs.

  (b) After prototype development, large costs of operationalization or a lack of research maturity may limit adoption likelihood.

  (c) After operationalization of the technology, it becomes clear that the user community does not yet have the capacity to use the results of the new technology. Strategies to address this include continuous capacity building among the user community, continuous improvement of the technology, and technology transfer.

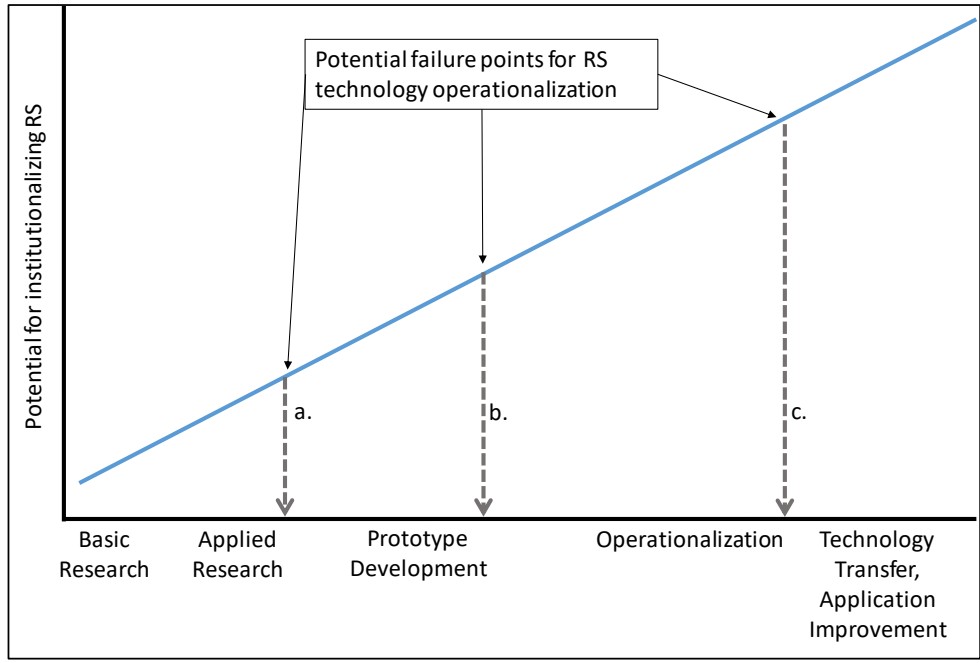

**Figure 4.** Conceptual graphic depicting the steps in a process in which the use of a new technology becomes institutionalized in Forest Inventory and Analysis (FIA). There are several points at which adoption of the technology might fail, including after applied research has been attempted (**a**), after prototype development (**b**), or after operationalization (**c**).

This case study reveals that FIA has created an environment where the competition of ideas and a culture of collegiality has led to creative thought and improvements in program efficiency, as well as a strong foundation of institutional knowledge that serves as a platform from which future research will advance.

## 4. Future Directions of RS Technology in FIA

In this section, several likely uses of new RS technology that could lead to increased operational efficiency in FIA or contribute to other advancements in forest monitoring are described.

### 4.1. RS Imagery Time Series

Use of FIA data in NAFD, LCMS, LCMAP, and more foundational work in LTS has set a clear course toward its future operationalization in FIA. Enhanced and novel LTS-based change detection algorithms are being explored in the FIA RS community (e.g., [256]), and opportunities for using data from other sensors, such as those from the European Union's Sentinel mission, with FIA data are in the knowledge discovery phase [162]. Analysis-ready LTS data are being created with new processing tools, making them more usable for operational monitoring [257]. It is likely that adopting LTS products will result in an increase of time series data use in mapping not only of change, but also of other FIA attributes (e.g., Wilson et al. [165]), and this will only become easier as more cloud-based options for computing and storage emerge.

Finally, in addition to LTS mapping applications, sample-based LTS approaches will be important in the future. For example, the current version of TimeSync [155] generates LTS graphics to aid staff performing sample-based PI, and Lister and Leites [162] showed how time series data associated with sample points can be used with machine learning to accurately identify forest cover change. Adopting a sample-based paradigm paves the way for its use to pre-screen inventory plots, which especially makes FIA's ICE project more efficient by identifying only the subset of FIA plots that need interpreting between ground visits. Moisen et al. [258] found that in order to report on land use and land cover trends in north central Georgia with adequate precision and temporal coherence, data needed to be collected on all the FIA plots each year over a long time series and broadly collapsed LULC classes. Relying heavily on TimeSync concepts, they call for a new, harmonized data collection approach that integrates ground, Landsat, and aerial imagery via a single enhanced plot interpretation process. Other FIA applications of sample-based LTS data include its potential for use to impute missing data at FIA plot locations that are unable to be measured in the field [259].

There are inherent advantages to using time series data as a sampling tool as opposed to a wall-to-wall mapping tool; maps require large amounts of additional resources in terms of storage, processing, and time spent on cartography and correcting visual discontinuities associated with image seamlines, clouds, and shadows. However, LTS data associated with sample points are more flexible, manageable, and can be linked to FIA plots for near real-time updates without the overhead of producing maps. Many natural resource monitoring programs around the world are turning to manual sample-based methods for estimating forest cover change, and many new efficient ocular PI tools have been developed [260].

### 4.2. Cloud Computing and Storage

Cloud and super-computing will clearly play a stronger role in the future of RS image processing, particularly when working with LTS or large volumes of high resolution imagery. As experienced with LandTrendR [170] and LCMS [261], transitioning to cloud computing in GEE opened possibilities for large area mapping that were previously not feasible. Similarly, supercomputers helped produce NAFD products [139]. GEE-based LTS advances include methods coming out of the Global Land Cover mapping and Estimation (GLanCE) project, which seeks to use GEE with the CCDC algorithm [171] to map land cover and land cover change annually at the global scale [262], as well as the recent addition

of NAIP to GEE, which will open up the potential for large area, high resolution pixel- or OBIA-based mapping using NAIP imagery every few years.

There are several new open source and commercial cloud-based RS environments and software tools being developed in addition to GEE [263]. For example, FIA's BIGMAP project [264] is based on a cloud architecture that includes Amazon Web Services, Esri's ArcGIS Enterprise Image Services, and ArcGIS applications, and seeks to leverage imagery and FIA plot data for improved estimation, mapping, data analysis and data distribution to FIA staff and clientele. The advantage of the BIGMAP project is that it is integrated with the Esri software ecosystem, in principle making it easier for GIS specialists to use. It also allows for the use of FIA's plot coordinate data in a corporate environment while keeping private information protected, exploitation of cloud-computing, and integration with various statistical applications and data types.

### 4.3. Exploitation of the Z-Dimension

The "Z-dimension" is a colloquial term that refers to data that confer information on structure above the ground, as opposed to that from the X-Y plane, which is typically associated with optical RS data. Airborne and spaceborne lidar, and advances in the use of radar, will provide new opportunities to increase FIA's efficiency and improve mapping quality.

#### 4.3.1. Airborne Lidar

Airborne lidar will likely occupy a growing role in FIA. Its unique value to FIA for work in remote areas of Alaska has been demonstrated [265], and as with LTS, it will become feasible for large area mapping as cloud-based technology improves and data discovery and distribution are standardized [215,266]. Furthermore, the USFS is currently conducting tests on lidar sensor systems that are combined with camera systems that could be used on future NAIP missions [267]; this would bring costs down, increase standardization, and shorten lidar revisit cycles, which at current use do not meet FIA's monitoring requirements. Large area lidar datasets could serve as covariates in predictive models of FIA attributes and forest cover change, and could serve as a means by which to help better-identify trees outside of forest and expand FIA tree data collection across all land types.

#### 4.3.2. Spaceborne Lidar

Structural information from ICESat and GEDI lidar systems could take on added importance as data become available and FIA's understanding of the data grows. However, a barrier to its adoption is the lack of understanding of how to integrate sample-based lidar that only has limited spatial overlap with FIA plots. New estimation approaches that combine field plots, lidar point sample locations, and wall-to-wall lidar or other RS data will be needed, along with appropriate estimators that produce results compatible with FIA estimates [246,247]. Efforts are currently underway to create a web-based estimation system that does just that: The Online Biomass Inference Using Waveforms and Inventory (OBI-WAN) project, which seeks to provide a carbon reporting system that uses GEDI, FIA data, and novel estimators [246] to produce reports for user-specified areas. Similar approaches could be adopted with the ICESat2 data that became available in 2020 [268]; however, it should be noted that both GEDI and ICESat2 have finite mission lives, and there is no guarantee of data stream continuity into the future.

#### 4.3.3. Radar

FIA has not conducted much research using radar data. A notable early application of radar exploits the InSAR (Interferometric Synthetic Aperture Radar) system carried on the Shuttle Radar Topography Mission (SRTM). With data from FIA plots, inSAR, and other imagery and GIS layers, Walker et al. [269] and Kellndorfer et al. [270] produced vegetation height maps for large sections of Utah and several northeastern states, respectively. Using FIA plot data, InSAR, and data from other sensors, Yu et al. [271] produced not only forest height but also biomass maps for the state of

Maine. Kellndorfer et al. [272] describe various realized and potential applications of using radar for large area mapping. Future work with SAR data for large area mapping will likely be conducted using cloud-based processing. For example, Google Earth Engine now hosts a long time series of Sentinel-1 SAR data, as well as algorithms for conducting change analyses with these [273]. As radar processing becomes more streamlined, such as in ways identified by resources such as Flores et al. [274], FIA scientists will be more likely to more fully adopt this promising technology.

*4.4. Improved Estimation*

A primitive, though unrealized vision for operationalization of a nascent form of SAE was presented by McRoberts and Wendt [275]. They advocated an approach where FIA clientele could identify arbitrary polygonal areas within-which FIA estimates could be generated using post-stratification information aggregated to cells in a tessellation of the country. In an unrelated effort, Proctor et al. [276] describe a well-known carbon tool, the Carbon OnLine Estimator (COLE), which did allow users to identify custom areas for which FIA-based carbon estimates were produced (also see Van Deusen and Heath [277]. After running for over 15 years, the COLE tool is currently not available as it is being updated for a cloud computing environment. More recently, new RS technology and increased computing resources allow FIA the opportunity to greatly improve estimation efficiency moving forward. A significant step toward operationalizing the use of alternative model-assisted inference is the development of the Forest Inventory Estimation for Analysis (FIESTA) software (Frescino et al., 2012), which currently has a model-assisted module that accesses the seven model-assisted estimators used in the statistical package mase [278]. FIESTA allows for not only the use of traditional FIA estimators that are compatible with corporate systems, but also the integration of estimators and data from machine learning for comparison with standard methods.

Moving beyond model-assisted inference, there is an increased need for small area estimates to characterize forest attributes within small domains such as counties, ecological boundaries, disturbance events, and shortened time intervals (i.e., SAE). Efforts to operationalize SAE have generally been hindered by a combination of technical challenges and reconciliation with FIA's standardized, peer-reviewed estimation system [14]. However, recently software developments and improvements in knowledge of machine learning and model-based inference have led to the development of new tools and demonstration projects. For example, the FIESTA software mentioned above also takes a step toward operationalizing the use of SAE methods by offering a module that easily links FIA and remotely sensed data through packages in the R software [279], including sae [280] and JoSAE [281]. Future developments in this area include a revision of FIA's estimation procedure handbook [14] to include SAE, improved model-based inference (e.g., Saarela et al. [246]), and tool creation to streamline adoption of novel techniques as they are developed and their value as part of FIA's standard operating procedures is established.

One possible step toward improving estimation in a way similar to the functionality of the COLE tool is the creation, through advanced k-nn techniques, of a national, pixel-based map of FIA plot identifier in the aforementioned BIGMAP platform that is under development. This national map, which would be driven in part by LTS data that are constantly refreshed, would serve as the foundation for summarization of FIA plot data at the pixel-level, allowing for the drawing of arbitrary polygonal areas and the calculation of estimates using either model-assisted or model-based inference [42]. A similar approach to this was already implemented by Wilson et al. [122] using MODIS data, and a Landsat-based FIA plot identifier map has been produced for the entire country for one time period using the technique described in Riley et al. [282]. The BIGMAP platform has the advantage that it can easily integrate with other FIA estimation infrastructure and databases, and thus potentially serve as a foundational platform for FIA's future operational use of SAE. Another possible step forward is to integrate FIESTA's statistical estimation capabilities with BIGMAP's powerful spatial toolsets to access a multitude of SAE tools already available on the Comprehensive R Archival Network (CRAN). Investigations are currently underway to identify best models and delivery systems.

## 5. Conclusions

FIA has a long history of using RS imagery and technology to improve efficiency. Beginning with hand-drawn maps, paper aerial photos, and coarse-resolution satellite imagery, FIA's use of RS has tracked advancements in technology, as it has tracked the progression of inventory data needs from solely-timber focused to more broadly ecological and social. It has evolved to exploit time series of satellite data such as Landsat, MODIS and Sentinel, and has been turning toward technologies that are linked to Z-dimensional structural information (height), such as lidar, stereo NAIP-derived point clouds, and radar. Technology advancements, including software, hardware, and networking capabilities, have made machine learning and cloud computing foundational for FIA's future RS advances, and led to a blossoming of efficiency-creating methods.

Barriers to operationalizing new technologies still exist, and include balancing research and core production functions, limited capacity and high costs to use new tools and data types, the need to maintain long-term continuity of results even as methods and outputs evolve, and use of production workflows that are difficult to change due to their complexity. However, expectations from stakeholders continue to encourage increased RS technology development. In addition, FIA has created an environment where the competition of ideas and a culture of collegiality has led to creative thought and improvements in program efficiency, as well as a strong foundation of institutional knowledge that serves as a platform from which future research will advance. NFIs such as FIA require a commitment to science, public service, and stewardship of institutional knowledge to evolve along with advancements in RS and natural resource monitoring science, while ensuring long-term comparable results.

**Author Contributions:** Conceptualization, A.J.L. and L.S.H.; writing—original draft preparation, A.J.L.; writing—review and editing, A.J.L., H.A., T.F., D.G., S.H., L.S.H., G.C.L., R.M., G.G.M., M.N., R.R., K.S., T.A.S., J.W., and B.T.W.; visualization, A.J.L.; project administration, A.J.L. All authors have read and agreed to the published version of the manuscript.

**Funding:** This research received no external funding.

**Acknowledgments:** This work was supported by the U.S. Department of Agriculture, Forest Service. The authors thank the contributions of a multitude of FIA staff and partners who have contributed to the research described herein.

**Conflicts of Interest:** The authors declare no conflict of interest.

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
