# Peer review of "Use of Remote Sensing Data to Improve the Efficiency of National Forest Inventories: A Case Study from the United States National Forest Inventory"

_forests, doi:10.3390/f11121364_

Round 1

Reviewer 1 Report

Presented paper - review- deals with using of remote sensing (RS) by National forest inventories (NFI) on example of United States of America. Authors presents Forest Inventory and Analysis program (FIA). Using of RS by NFI has more irretrievable benefits. 

Review is designed in five Chapters. The first is Introduction, which in 7 pages presents Values and uses of methods and models in NFI. The Second Chapter has largest extent with 12 pages, and presents progression of FIA from their starting to present-day. In details are presented and discussed used methods from photointerpretation, per radiometer, Landsat, MODIS, cloud computing, to LIDAR and other modern systems. 

The third Chapter briefly presents general observations on RS data integration in FIA and other NFIs. The fourth Chapter suggest future direction of RS technology in FIA and last is the fifth chapter, which briefly on half page make conclusions of this Review.

The Review is supplemented very comprehensive references on 20 pages with nearly 300 mentioned papers from this research area.

I consider whole Review as very good prepared on high scientific level, clear and comprehensive. I recommend public this Review in presented form. I have only little remark, in row 1346 is by my opinion clerical error by AVHHR (not AVHRR?)

Author Response

Review Report Form; Reviewer 1

Comments and Suggestions for Authors

Presented paper - review- deals with using of remote sensing (RS) by National forest inventories (NFI) on example of United States of America. Authors presents Forest Inventory and Analysis program (FIA). Using of RS by NFI has more irretrievable benefits.

Review is designed in five Chapters. The first is Introduction, which in 7 pages presents Values and uses of methods and models in NFI. The Second Chapter has largest extent with 12 pages, and presents progression of FIA from their starting to present-day. In details are presented and discussed used methods from photointerpretation, per radiometer, Landsat, MODIS, cloud computing, to LIDAR and other modern systems.

The third Chapter briefly presents general observations on RS data integration in FIA and other NFIs. The fourth Chapter suggest future direction of RS technology in FIA and last is the fifth chapter, which briefly on half page make conclusions of this Review.

The Review is supplemented very comprehensive references on 20 pages with nearly 300 mentioned papers from this research area.

I consider whole Review as very good prepared on high scientific level, clear and comprehensive. I recommend public this Review in presented form. I have only little remark, in row 1346 is by my opinion clerical error by AVHHR (not AVHRR?)

RESPONSE: Thank you for your kind words. We have made the edit you suggest.

In addition to the above-mentioned revisions, we updated figure 2 and 3, made some small edits to text, and reformatted the references.

Reviewer 2 Report

The review presents the history of the use of Remote Sensing (RS) in the National Forest Inventory (NFI) of the United States and describes present day applications and future directions of this technique in NFI programs.

I think the review meets the goal to offer Forest Inventory and Analysis (FIA)’s experience with NFI-RS integration so it can be very useful for other countries that want to improve the efficiency of their NFI programs.

Requested revisions:

  • General background on FIA and sampling statistics are included, but I miss a brief description of NFI of United States (number of plots, areas, variables, etc…) in the introduction section.
  • Line 136: Reference.
  • Line 142: Reference
  • Line 228: Equal sign is missed.
  • Line 272: Do you mean area-level?
  • Lines 277-279: Reference.
  • Line 938: Do you mean UAS?

Author Response

Review Report Form; Reviewer 2

Comments and Suggestions for Authors

The review presents the history of the use of Remote Sensing (RS) in the National Forest Inventory (NFI) of the United States and describes present day applications and future directions of this technique in NFI programs.

I think the review meets the goal to offer Forest Inventory and Analysis (FIA)’s experience with NFI-RS integration so it can be very useful for other countries that want to improve the efficiency of their NFI programs.

Requested revisions:

  • General background on FIA and sampling statistics are included, but I miss a brief description of NFI of United States (number of plots, areas, variables, etc…) in the introduction section.

RESPONSE: We added a brief background on the NFI of the US, in the first paragraph of section 1.2 (below line 88).

  • Line 136: Reference.
  • Line 142: Reference
  • Line 228: Equal sign is missed.
  • Line 272: Do you mean area-level?
  • Lines 277-279: Reference.
  • Line 938: Do you mean UAS?

RESPONSE: We made the indicated suggestions by adding references and clarifying the text.

In addition to the above-mentioned revisions, we updated figure 2 and 3, made some small edits to text, and reformatted the references.

Reviewer 3 Report

Thank you for this review of how FIA has incorporated remote sensing technologies into the National Forest Inventories over time. I think this paper will be a useful contribution to staff working on National Forest Inventories in other countries.

Author Response

Review Report Form; Reviewer3             

Comments and Suggestions for Authors

Thank you for this review of how FIA has incorporated remote sensing technologies into the National Forest Inventories over time. I think this paper will be a useful contribution to staff working on National Forest Inventories in other countries.

RESPONSE: Thank you for your kind comments.

In addition to our revisions from other reviewers, we updated figure 2 and 3, made some small edits to text, and reformatted the references.